# Immunogenicity of a 20-Valent Pneumococcal Conjugate Vaccine Versus a 13-Valent Vaccine in Infants: A Systematic Review and Meta-Analysis

**DOI:** 10.3390/vaccines13111156

**Published:** 2025-11-12

**Authors:** María-Dolores Pacheco-Haro, Sergio Núñez de Arenas-Arroyo, Valentina Díaz-Goñi, Elisa-Janeth Velasco-Lucio, Carol-Ingrid Castellares-González, Valeria Reynolds-Cortez, Adriana Simeón-Prieto, Elsa Ignateva, Vicente Martínez-Vizcaíno

**Affiliations:** 1Health and Social Research Centre, Universidad de Castilla-La Mancha, 16002 Cuenca, Spain; 2Preventive Medicine, Hospital Virgen de la Luz, 16002 Cuenca, Spain; 3Preventive Medicine, Hospital Universitario de Toledo, 45007 Toledo, Spain; 4Preventive Medicine, Hospital Universitario 12 de Octubre, 28041 Madrid, Spain; 5Facultad de Ciencias de la Salud, Universidad Autónoma de Chile, Talca 3460000, Chile

**Keywords:** immunogenicity, pneumococcal conjugate vaccines, PCV20, PCV13, 20-valent, 13-valent, meta-analysis, geometric means ratio, opsonophagocytic activity

## Abstract

Background/Objectives: The 20-valent pneumococcal conjugate vaccine (PCV20) was approved for use in children and infants on the basis of studies comparing its safety and immunogenicity with those of the 13-valent vaccine (PCV13). PCV20 offers expanded coverage of seven additional serotypes. This meta-analysis aimed to summarize the available evidence on the comparative immunogenicity between PCV20 and PCV13. Methods: A systematic search of the PubMed, Web of Science, Scopus, Cochrane, and ClinicalTrials.gov databases was conducted in September 2024. The following inclusion criteria were used: (i) design: randomized clinical trials; (ii) outcomes: studies that included immunogenicity outcomes; (iii) compared vaccines: any study directly comparing the immunogenicity of PCV20 and PCV13; and (iv) population: infant population <2 years of age. No language or temporal restrictions were applied in the study. A random-effects meta-analysis was conducted via the Hartung–Knapp–Sidik–Jonkman method, with subgroup analyses according to the serotype and vaccination schedule (3 + 1 and 2 + 1). We used the revised Cochrane risk of bias 2 tool (RoB 2.0) to assess the risk of bias. The following parameters of immunogenicity were estimated: (i) the pooled geometric mean ratio (GMR PCV20/PCV13) of serotype-specific pneumococcal anticapsular antibodies, (ii) the pooled difference (PCV20-PCV13) in the percentage (DP) of participants who achieved predefined antibody levels for each serotype, and (iii) the pooled geometric mean titres (GMTs) of serotype-specific opsonophagocytic activity (OPA) in PCV20 and PCV13, along with their 95% confidence intervals (95% CIs). Results: Four studies (4093 infants aged 42–180 days) that compared the PCV20 and PCV13 vaccines, published between 2021 and 2024, were included in this meta-analysis. The immunogenicity of both groups was compared one month after the primary series and one month after the booster dose. The pooled results indicated that PCV20 elicited lower immune responses for the 13 serotypes shared with PCV13, according to the GMR and OPA outcomes. For the DP outcome, no statistically significant differences were observed between the two groups. Immune responses were higher for the additional serotypes in the PCV20 group; however, these differences were not statistically significant for all serotypes. Conclusions: This meta-analysis offers an overview of the evidence on the comparative immunogenicity of PCV20 and PCV13. Although some outcomes indicate that PCV20 elicits lower immune responses for the 13 serotypes shared with PCV13, it provides immunity against seven additional serotypes associated with IPD. Further studies are warranted to strengthen the evidence base, and continuous IPD surveillance remains essential to monitor shifts in serotype prevalence, assess the impact of current and future vaccines, and guide vaccine policy recommendations.

## 1. Introduction

*Streptococcus pneumoniae* (*S. pneumoniae*) is an encapsulated Gram-positive bacterium that colonizes the mucosal surfaces of the human upper respiratory tract as part of the nasopharyngeal flora. Occasionally, this bacterium causes pneumococcal disease (PD) [1,2,3]. While noninvasive forms of PD, such as sinusitis and otitis media, are more prevalent, invasive forms (IPD), including pneumonia, meningitis, and bacteremia, are more severe [1,2,4]. In the general population, PD contributes significantly to morbidity and mortality from infectious diseases, particularly in children, older adults, and immunocompromised individuals [4,5,6,7,8,9,10]. According to the Global Burden of Diseases 2021 study [7], *S. pneumoniae* was the leading cause of lower respiratory tract infections and related deaths, with an estimated 97.9 million episodes and 505,000 deaths globally. It was also identified as one of the top five causes of deaths associated with antimicrobial resistance worldwide.

Pneumococcal conjugate vaccines (PCVs) play crucial roles in preventing bacterial respiratory infections caused by *S. pneumoniae* [4]. Since their introduction in 2000, they have contributed to a significant reduction in the global burden of PD and IPD in children [4,8,11,12,13,14,15,16,17,18,19]. This effect is attributable to their wide availability and incorporation into global immunization programs in accordance with World Health Organization (WHO) recommendations [3]. The polysaccharide capsule of *S. pneumoniae* is an essential virulence factor, and pneumococcal serotypes are defined based on differences in its composition [3]. To date, over 100 different pneumococcal serotypes have been described, although not all cause disease in humans [4]. Protection against PD by PCVs is achieved through the binding of serotype-specific antibodies to the pneumococcal polysaccharide capsule.

However, given that some serotypes not covered by current vaccines continue to cause significant disease, and that a considerable burden of IPD persists in children—with infants under one year of age being particularly vulnerable—there is an urgent need for higher valency pneumococcal vaccines covering an increasing number of *S. pneumoniae* serotypes [8]. For example, in Europe, an increasing proportion of IPD cases are caused by serotypes unique to PCV20, accounting for more than 61.8% of all IPD cases among children under one year of age in 2022 [20].

Recent technological advancements have had considerable impacts on the development of pneumococcal vaccines, resulting in the addition of new serotypes and broader coverage [8].

To guarantee the quality, safety, and efficacy of new PCVs, the WHO established minimum serological criteria for immunogenicity and noninferiority that must be met for approval [3,10,21,22,23,24]. The new PCV20 was authorized based on comparative data demonstrating noninferiority in terms of safety and immunogenicity to the 13-serotype pneumococcal conjugate vaccine (PCV13) [25]. It also offers advantages in terms of broader serotype coverage and a potential reduction in IPD by including seven serotypes causing pediatric IPD (8, 10A, 11A, 12F, 15B, 22F and 33F) [26,27,28,29,30,31,32,33]. According to several studies [34,35,36,37], differences in immunogenicity parameters between PCV20 and PCV13 have been reported. However, a meta-analysis has not yet systematically assessed these findings, leaving unresolved whether one vaccine is consistently more immunogenic.

Therefore, following the WHO immunogenicity criteria [3,10], this systematic review and meta-analysis aimed to determine whether these immunogenicity endpoints were greater for either vaccine: (i) serotype-specific immunoglobulin G geometric mean concentration (GMC), (ii) percentage of participants who achieved predefined antibody levels for each serotype, and (iii) serotype-specific functional antibody titres, measured at least 4 weeks after completion of the primary infant vaccination series.

## 2. Materials and Methods

Our systematic review with meta-analysis (PROSPERO registration number: CRD420251023915) was conducted following the Preferred Reporting Items for Systematic Reviews and Meta-Analyses (PRISMA-S and PRISMA) [38] guidelines (Figure 1) and according to the recommendations of the Cochrane Handbook for Systematic Reviews of Interventions [39].

### 2.1. Databases and Search Strategy

We searched the PubMed, WOS, Scopus, and Cochrane databases for randomized clinical trials in September 2024, with no language restrictions. To achieve maximum sensitivity, the search was performed by combining Medical Subject Headings (MeSH) terms with free terms in the title and abstract for pneumococcal conjugate vaccines in children: “child”, “infant”, “toddler”, “*Streptococcus pneumoniae*”, “conjugate pneumococcal vaccines”, “vaccines”, “valent”, “immunogenicity”, “immunoglobulin”, “IgG”, “antigenicity”, “PCV20”, “PCV13”, “13-valent”, “20-valent”, “randomized controlled trial”, “randomized controlled trials”, “randomized”, and “clinical trial”. In addition, we supplemented the electronic search with manual searches by reviewing the reference lists of previous reviews and with searches for additional published, unpublished, and ongoing randomized controlled trials (RCTs) in international trial registers such as ClinicalTrials.gov [40]. The detailed search strategy and terms for each database are presented in Appendix A of the Appendix A.

### 2.2. Eligibility Criteria

We included (i) randomized clinical trials (RCTs) published in any language, (ii) studies in infants <2 years of age, and (iii) studies directly comparing the immunogenicity of PCV20 and PCV13, providing estimates of antibody responses (serotype-specific pneumococcal IgG) at least one time point between 4 and 6 weeks after the primary vaccination series and/or 1 month after booster vaccination.

The exclusion criteria were as follows: (i) study design: any non-RCT design; (ii) outcomes: studies that did not include immunogenicity outcomes; (iii) compared vaccines: any study that did not directly compare PCV20 with PCV13; and (iv) population: studies in adults.

Two reviewers independently conducted the literature search, screening and trial selection (M-D.-P-H. and S.NDA-A.). A third researcher resolved disagreements (V. M.-V.).

### 2.3. Data Extraction and Outcome Definition

The following data were extracted from the original reports: (i) first author and year of publication, (ii) trial registration number, (iii) country of study, (iv) age of the study population, (v) number of participants and sex, (vi) vaccination schedule (e.g., two priming doses followed by a booster (2 + 1) or three priming doses followed by a booster (3 + 1), and (vii) outcomes assessed. Data were extracted independently by 2 reviewers (M-D.-P-H. and S.NDA-A.). In cases of disagreement, a third investigator made the final decision (V. M.-V.).

Our aim was to compare the immunogenic response of PCV20 with that of PCV13 as measured by (i) the PCV20/PCV13 geometric mean ratio (GMR) of serotype-specific anticapsular pneumococcal immunoglobulin G antibodies; (ii) the difference in the percentage (PCV20-PCV13) of participants who achieved predefined antibody levels for each serotype (DP); and (iii) the geometric mean titres (GMTs) of serotype-specific opsonophagocytic activity (OPA) in PCV20 and PCV13. Each patient was assessed 1 month after the last dose of the primary series and the booster dose. For the 13 matched serotypes, the PCV20 group was compared with the corresponding serotypes in the PCV13 group. For the seven additional serotypes, the PCV20 group was compared with the lowest result among the 13 serotypes in the PCV13 group (excluding serotype 3 because of its atypical immunogenicity). The predefined IgG concentrations were ≥0.35 μg/mL, except for serotypes 5 (≥0.23 μg/mL), 6B (≥0.10 μg/mL) and 19A (≥0.12 μg/mL) [3,10,41].

### 2.4. Risk of Bias Assessment

In accordance with the Cochrane Handbook for Systematic Reviews of Intervention, we used the revised Cochrane Risk of Bias 2 tool (RoB 2.0) [42] to assess the risk of bias according to five domains: (i) randomization process, (ii) deviations from intended interventions, (iii) missing outcome data, (iv) measurement of the outcome, and (v) selection of the reported result. Overall, we determined to have a “low risk of bias” if all domains were “low risk,” to have “some concerns”, if at least one domain was designated “some concerns” and no domains were identified as “high risk,” and to have a “high risk” if at least one domain was “high risk”.

The risk of bias assessment was performed independently by two reviewers (M-D.-P-H. and S.NDA-A.), and inconsistencies were resolved by consensus or by involving a third researcher (V. M.-V.).

### 2.5. Certainty of Evidence

We used the Grading of Recommendations Assessment, Development and Evaluation (GRADE) tool to judge the certainty of evidence. We judged the certainty of evidence for each outcome as high, moderate, low, or very low, on the basis of trial design, risk of bias, inconsistency, indirect evidence, imprecision, and publication bias.

### 2.6. Data Synthesis and Statistical Analysis

We conducted a random-effects meta-analysis via the Hartung–Knapp–Sidik–Jonkman [43,44] method. We conducted subgroup analyses according to the serotype and vaccination schedule employed (3 + 1 or 2 + 1) and estimated (i) pooled GMR, (ii) pooled DP, and (iii) pooled GMTs of OPA for PCV20 compared with those for PCV13 with 95% confidence intervals (95% CIs).

For these analyses, we assessed heterogeneity via the I2 statistic, and the results were categorized as follows: might not be important (0–40%), moderate heterogeneity (30–60%), high heterogeneity (50–90%), and very high heterogeneity (75–100%) [45]. The corresponding *p*-values were also considered [46]. Finally, we calculated the τ2 statistic to establish the size and clinical relevance of heterogeneity.

We performed a sensitivity analysis via the leave-one-out method [45] to assess the robustness of the summary estimates (Appendix A).

We conducted all analyses in R (version 4.2.2) [47] via the R package “meta”.

## 3. Results

### 3.1. Systematic Review

After removing duplicates, 452 citations were identified. Among them, 18 were considered potentially eligible and were examined in full text. Finally, we included four RCTs [35,36,37,38] (Figure 1), involving 4093 infants aged 42 days to 180 days, 2006 (49%) of whom were female. These trials were published between 2021 and 2024 and were conducted in countries in Europe, Australia, the United States/Puerto Rico, and Japan. The clinical trials were randomized into two groups to receive PCV20 (2051) or PCV13 (2042), and the immunogenicity of both groups was compared one month after the primary series and one month after the booster dose. Three studies used a 3 + 1 [36,37,38] dose vaccination schedule, and one study [35] used 2 + 1 doses. The main characteristics of the four included studies are shown in Table 1. The reasons for the exclusion of studies are available in Appendix A of the Appendix A.

### 3.2. Results of Risk of Bias

The risk of bias of the studies was assessed via the ROB 2 tool [42], which revealed a low risk of bias. The Appendix A contain a comprehensive report that addresses each question of the RoB 2.0 tool from each of the included studies for each outcome.

### 3.3. Meta-Analysis

All four studies [35,36,37,38] reported the immune response as GMR or DP after the primary series. Figure 2 and Figure 3, respectively, show pooled effects obtained in the meta-analysis via the random-effects model. The certainty of the evidence is presented in Appendix A.

As shown in Figure 2, there was a statistically significant reduction in the immunogenicity of PCV20 relative to that of PCV13 for all the shared serotypes. Among the additional serotypes, PCV20 had greater effects than PCV13 did for all seven serotypes, although these effects reached statistical significance only for serotypes 15B, 22F and 33F. The certainty of the evidence was high for GMR results after the primary dose in shared serotypes but low for additional unique serotypes of PCV20.

Figure 3 shows a decrease in the percentage of participants who achieved the predefined GMC for shared serotypes when PCV20 was used compared with PCV13, which was statistically significant only for serotype 3. For additional serotypes, the results revealed a greater percentage for PCV20 for serotypes 8, 11A, 15B, 22F and 33F and a lower percentage for serotypes 10A and 12F. However, these results did not achieve statistical significance in either case. The certainty of the evidence ranged from low to high for shared serotypes, depending on the serotype, and from low to very low for additional serotypes unique to PCV20.

Additionally, all four studies [35,36,37,38] reported immune responses as GMRs after the booster dose (Figure 4), and two [35,36] reported DP (Figure 5).

As shown in Figure 4, the immunogenicity of PCV20 was significantly lower than that of PCV13 for all shared serotypes. For the additional serotypes, immunogenicity was higher in the PCV20 group for all serotypes (8, 10A, 11A, 15B, 22F and 33F), except for 12F. The results showed statistical significance for all serotypes except for 8. The certainty of the evidence was high for shared serotypes, but low to very low for additional unique serotypes of PCV20.

For the DP outcome, Figure 5 shows that no statistically significant differences were observed between the two groups for the shared serotypes. Only the results for serotypes 3 and 23F were statistically significant. For serotypes unique to PCV20, the results indicated that for serotypes 8, 10A, 11A, 15B, 22F, and 33F, the percentage of participants who exceeded the prespecified GMC levels was higher in the PCV20 group than in the PCV13 group. Only for serotype 12F was the value higher for PCV13. Statistically significant differences were observed for serotypes 8 and 10A. Overall, the comparison between PCV20 and PCV13 revealed minimal differences. For all serotypes, the DP outcome showed a high level of evidence after the booster dose.

All four studies [35,36,37,38] evaluated the OPA GMTs one month after the primary series and after the booster dose. Figure 6 and Figure 7 present the pooled results from the 3 + 1 vaccination studies. Appendix A in the Appendix A show the results of the 2 + 1 vaccination schedule study.

After the primary series (Figure 6), elevated pooled GMTs were observed for serotypes shared by both vaccines in the PCV20 and PCV13 groups. Overall, OPA GMTs were higher in the PCV13 group than in the PCV20 group, indicating a greater opsonophagocytic capacity of antibodies induced with PCV13. Only the results for serotypes 4 and 7F were higher in the PCV20 group than in the PCV13 group. For the additional serotypes, OPA GMTs in the PCV20 group were significantly higher than those in the PCV13 group. The certainty of the evidence for OPA in the PCV20 group was low to very low across all serotypes, whereas in the PCV13 group it ranged from very low to high, depending on the serotype (Appendix A).

After the booster dose was administered (Figure 7), a similar trend was observed. For the shared serotypes, OPA GMTs were lower for PCV20 than for PCV13, whereas they were higher for the additional serotypes. As with the OPA results after the primary series, the certainty of the evidence for OPA in the PCV20 group was low to very low across all serotypes. In the PCV13 group, the certainty of the evidence ranged from very low to high, depending on the serotype.

The results of the subgroup analysis according to vaccination schedule type (3 + 1 or 2 + 1) for each serotype are presented in Appendix A. The immunogenicity results derived from the subgroup analysis indicate that the 3 + 1 schedule elicited a more robust immune response than the 2 + 1 schedule for all serotypes, both after the primary series and following the booster dose. However, the available evidence is limited, as the analysis included only a single study using the 2 + 1 schedule.

## 4. Discussion

This meta-analysis synthesizes the available evidence on the immunogenicity of PCV20 compared with that of PCV13 in healthy infants. These findings indicate that PCV20 exhibits reduced immunogenicity for the 13 serotypes shared with PCV13, based on the analyzed outcomes, except for the DP outcome, for which no statistically significant differences were observed between the two groups. However, PCV20 confers immunogenicity against seven additional serotypes (8, 10A, 11A, 12F, 15B, 22F, and 33F), which are not covered by PCV13 and are increasingly associated with IPD [29,31,48,49,50,51].

Several systematic reviews and meta-analyses have reported results concerning pneumococcal conjugate vaccines but not immunogenicity [15,19,29,52,53,54,55,56,57,58,59,60,61,62,63,64,65,66,67,68]; others have focused on the immunogenicity of pneumococcal conjugate vaccines in adults but not in infants [68,69,70,71,72,73,74]. Our meta-analysis is the first to evaluate the comparative immunogenicity between PCV20 and PCV13 in healthy infants.

Although PCV20 showed reduced immunogenicity for the shared serotypes in the outcomes of the GMR and OPA in this meta-analysis, it met the WHO criteria for noninferiority (the percentage of PCV recipients with serotype-specific immunoglobulin G ≥ 0.35 μg/mL and the serotype-specific IgG geometric mean concentration after vaccination) [3,10,25,75]. It remains unclear whether a lower serotype-specific GMC of antibodies correlates with reduced efficacy for those serotypes. The threshold is intended to establish noninferiority against the reference PCV in aggregate, and no serotype-specific thresholds have been defined [3]. Nevertheless, noninferiority does not imply clinical equivalence. The sequential application of noninferiority criteria may have cumulative negative consequences, and even modest reductions in antibody responses could result in diminished protection over time, particularly in populations with lower vaccine responsiveness or population suboptimal coverage [76]. Future studies should determine whether these differences in immunogenicity translate into differences in vaccine effectiveness, especially with respect to protection against IPD [77].

Another significant concern that has emerged is serotype replacement by nonvaccine strains, leading to an increased incidence of IPD [78]. While PCV20 has expanded serotype coverage, continuous IPD surveillance remains essential to monitor shifts in serotype prevalence, assess the impact of current and future vaccines, and guide vaccine recommendations [29,79].

Compared with the 2 + 1 schedule, the 3 + 1 vaccination schedule has been shown to elicit higher OPA responses. However, some findings [80] indicate that these differences do not reduce vaccine efficacy against IPD in children. Further reductions in the number of childhood pneumococcal vaccines, such as the 1 + 1 schedule implemented in the United Kingdom, may ultimately result in reduced effectiveness. Additional research is needed to determine the overall superiority of each schedule. When formulating a schedule, it is essential to consider factors such as the country’s income level, prevalent serotypes, incidence of pneumococcal disease, and cost-effectiveness analyses.

### 4.1. Implications for Practice and Policy

While the use of PCV20 appears justified based on WHO criteria and broader serotype coverage, an antibody titer that reliably correlates with protection against IPD caused by any single serotype is unknown [77]. Linking immunogenicity and effectiveness is necessary to assess whether changes in the immune response from currently recommended PCVs to next-generation vaccines could impact overall effectiveness [81]. National immunization programs should therefore consider the potential trade-off between reduced immunogenicity for shared serotypes and expanded protection against emerging serotypes. Epidemiological surveillance of circulating serotypes, along with effectiveness and cost-effectiveness analyses, should be integrated with immunogenicity findings to inform public health decisions, particularly in low- and middle-income countries where disease burden and vaccine affordability vary substantially [82].

Future research should focus on (1) a thorough examination of the vaccine’s impact, duration of protective efficacy, and the indirect effects of different dosing schedules; (2) analysis of serotype replacement; and (3) further establishment of serotype-specific immune correlates of protection against invasive pneumococcal disease (IPD) in diverse transmission settings.

### 4.2. Limitations

The results of our meta-analysis should be interpreted with caution due to several limitations. First, the number of included studies was limited, and all were funded by the manufacturer of PCV20. Second, although the clinical trials included in this review were of high methodological quality, substantial heterogeneity was observed for certain outcomes. No factors were identified that could explain the variation in results, although one possible source of heterogeneity is the use of different antibody testing procedures across studies. Additionally, we were unable to assess publication bias due to the limited number of studies available. Another limitation is the variation in vaccination schedules: one study used a 2 + 1 schedule, whereas the other three used a 3 + 1 schedule. Therefore, a schedule with only two primary doses may result in different immunogenicity after the booster dose compared with three primary doses.

## 5. Conclusions

This systematic review and meta-analysis provide the first comprehensive synthesis of comparative immunogenicity data between PCV20 and PCV13 in healthy infants. Although PCV20 elicited lower immune responses for the 13 shared serotypes in some outcomes, it fulfilled the WHO noninferiority criteria, indicating that it meets the minimum immunological thresholds required for licensure. In addition, PCV20 induced immune responses to seven additional serotypes not included in PCV13, which are increasingly associated with invasive pneumococcal disease. These findings suggest that broader serotype coverage could offer significant public health benefits, particularly in settings where emerging non-PCV13 serotypes contribute to disease burden.

However, noninferiority in immunogenicity does not necessarily translate into equivalent clinical protection, especially in the absence of real-world effectiveness data. While PCV20 appears to be a promising next-generation pneumococcal conjugate vaccine, its true impact on PD prevention and population-level outcomes remains to be established. Ongoing surveillance, as well as effectiveness and health-economic evaluations, will be crucial to assess the cost-effectiveness of PCV20 in diverse epidemiological contexts.

## Figures and Tables

**Figure 1 vaccines-13-01156-f001:**
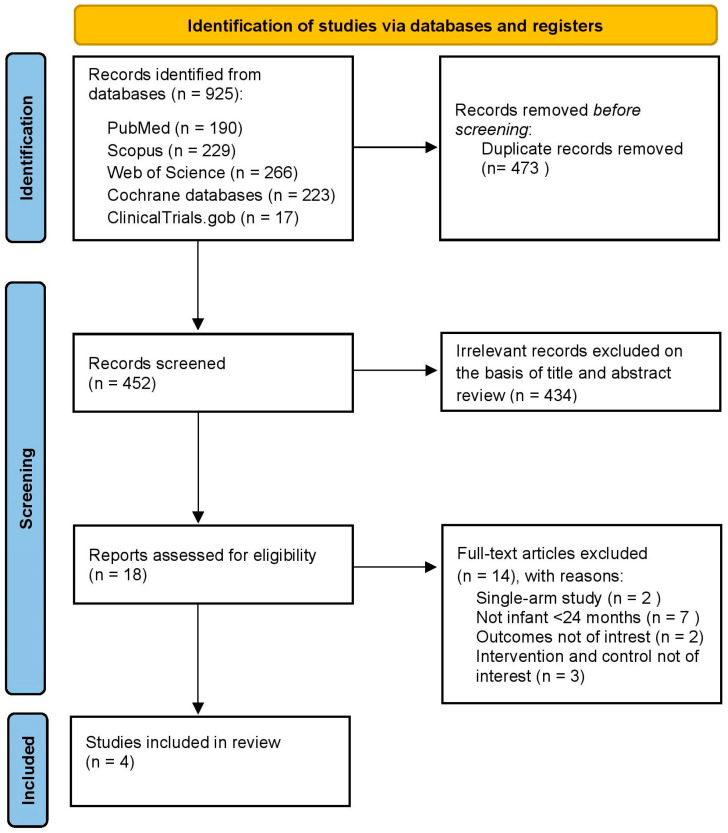
PRISMA flowchart.

**Figure 2 vaccines-13-01156-f002:**
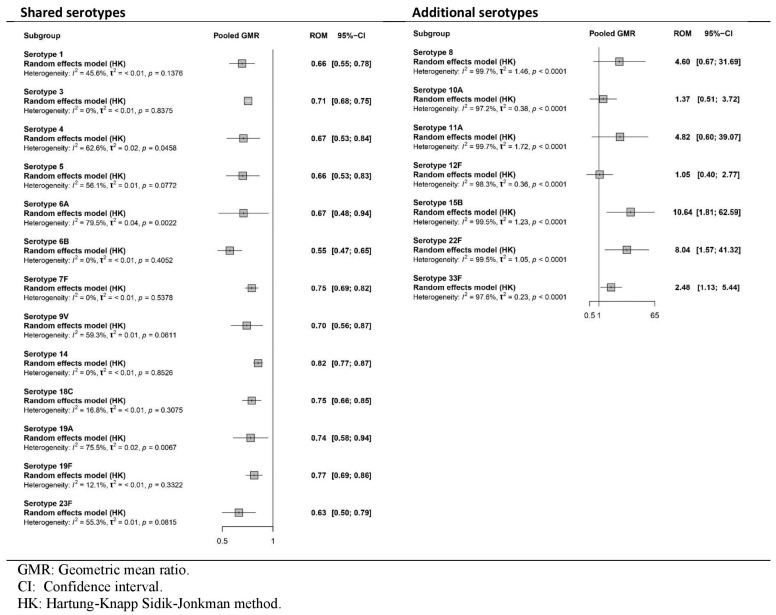
Meta-analysis of the GMR (PCV20/PCV13) of serotype-specific anti-capsular pneumococcal immunoglobulin G and 2-sided 95% CIs after primary series vaccination (forest plot).

**Figure 3 vaccines-13-01156-f003:**
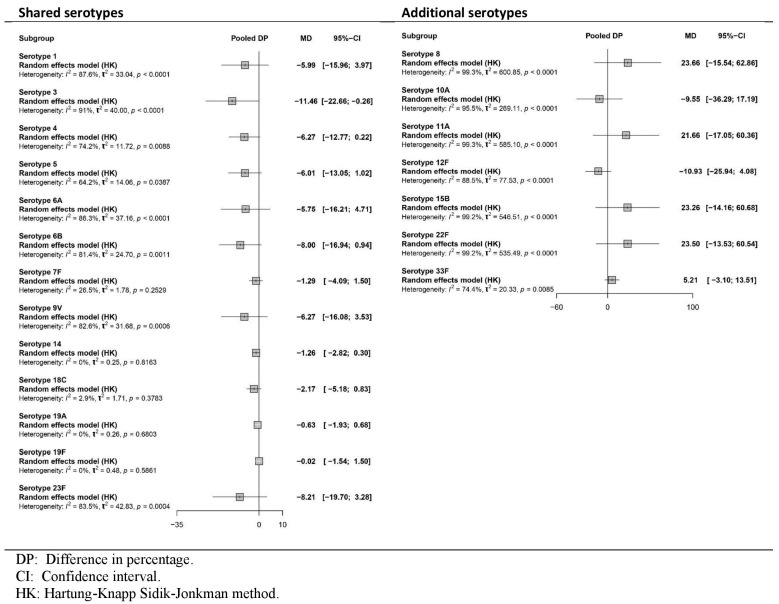
Meta-analysis of the difference (PCV20-PCV13) in the percentage (DP) of participants achieving predefined antibody levels for each serotype after the primary series vaccine (forest plot).

**Figure 4 vaccines-13-01156-f004:**
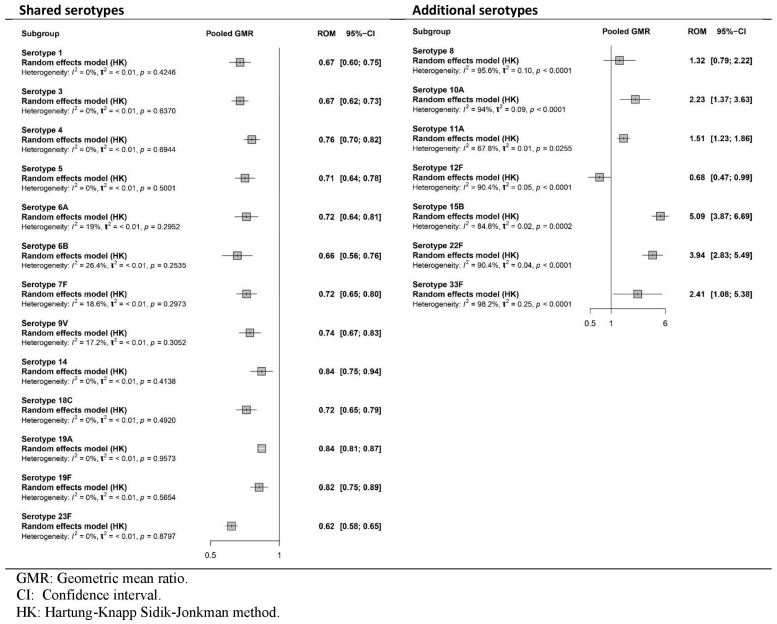
Meta-analysis of the GMR (PCV20/PCV13) of serotype-specific anticapsular pneumococcal immunoglobulin G and 2-sided 95%. CIs after booster vaccination (forest plot).

**Figure 5 vaccines-13-01156-f005:**
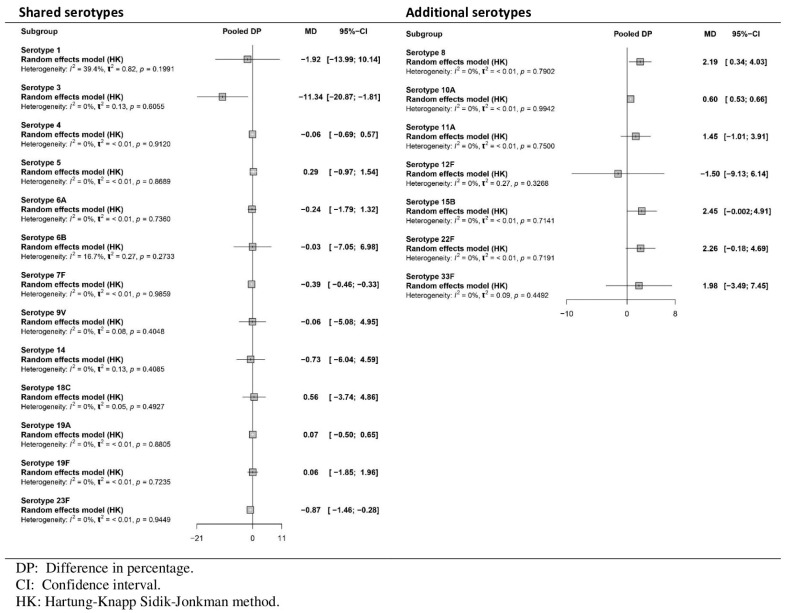
Meta-analysis of the difference (PCV20-PCV13) in the percentage of participants who achieved predefined antibody levels for each serotype after booster vaccination (forest plot).

**Figure 6 vaccines-13-01156-f006:**
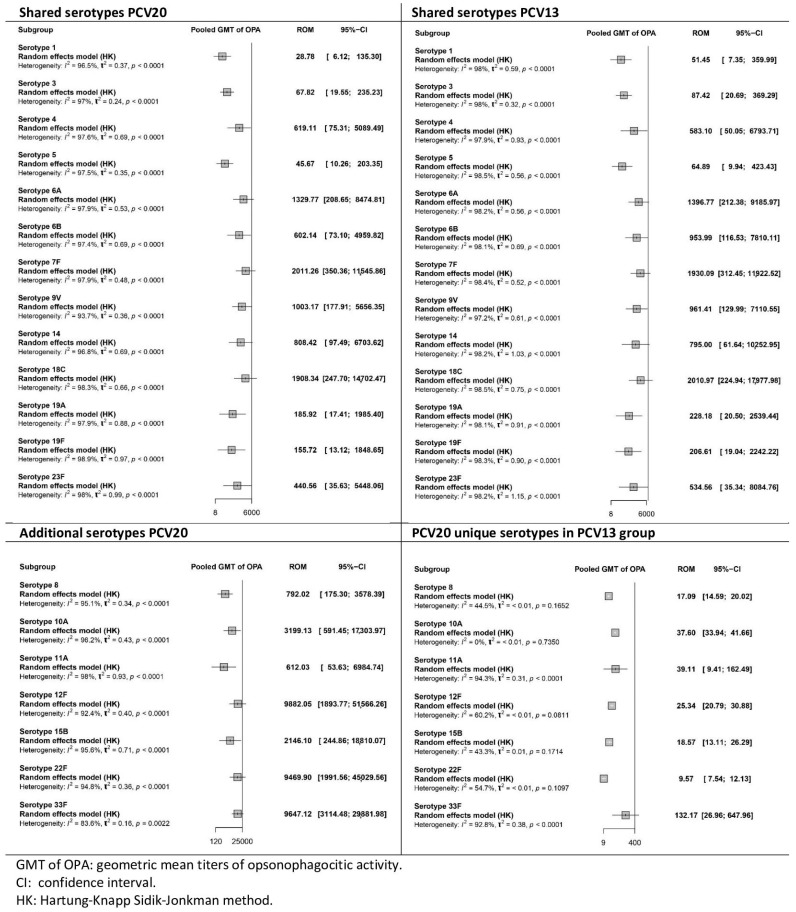
Meta-analysis of the geometric mean titres of serotype-specific opsonophagocytic activity after primary series vaccination with PCV20 and PCV13 (forest plot).

**Figure 7 vaccines-13-01156-f007:**
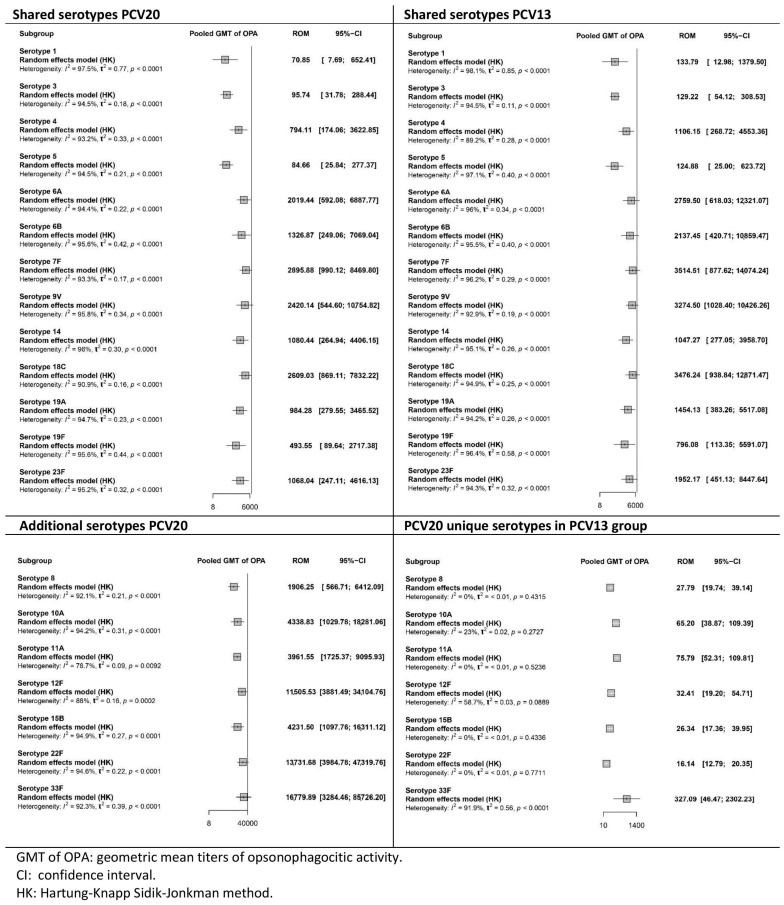
Meta-analysis of the geometric mean titres of serotype-specific opsonophagocytic activity after booster dose vaccination with PCV20 and PCV13 (forest plot).

**Table 1 vaccines-13-01156-t001:** Characteristics of the included studies.

Author and Publication Year	Study Design	Country	Population (Age in Days)	N (% Female)	Schedule	Schedule Primary Series	Schedule Booster	Evaluated Outcomes *	Risk of Bias
Korbal, 2024 [34]	RCT NCT04546425	Europe and Australia	Healthy infants (42–112)	IG: 601 (50.2) CG: 603 (48.4)	2 + 1	2–3, 4–5 months	11–12 months	(1, 2, 3, 4)	Low
Senders, 2024 [35]	RCT NCT04382326	United States/ Puerto Rico	Healthy infants (42–98)	IG: 1001 (48.3) CG: 987 (48.8)	3 + 1	2, 4, 6 months	12–15 months	(1, 2, 3, 4)	Low
Ishihara, 2024 [36]	RCT NCT04530838	Japan	Healthy infants (60–180)	IG: 217 (48.8) CG: 224 (50.9)	3 + 1	**	**	(1, 2, 4)	Low
Senders, 2021 [37]	RCT NCT03512288	United States	Healthy infants (42–98)	IG: 232 (48.3) CG: 228 (50.4)	3 + 1	2–6, 4, 6 months	12 months	(1, 2, 4)	Low

* Immunogenicity outcomes: (1) Geometric means ratio (PCV20/PCV13) of serotype-specific anti-capsular pneumococcal immunoglobulin G, after primary series and booster dose. (2) Difference (PCV20-PCV13) in the percentage of participants achieving predefined antibody levels for each serotype, after primary series. (3) Difference (PCV20 - PCV13) in the percentage of participants achieving predefined antibody levels for each serotype, after booster dose. (4) Geometric mean titres of serotype-specific opsonophagocytic activity after primary series and booster dose vaccines. ** Dose 1 at 2–6 months of age, doses 2 and 3 at 4- to 8-week intervals. RCT: Randomized controlled trials. IG: Intervention group. CG: Control group. Risk of bias was assessed based on Cochrane Risk of Bias tool.

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
