# Peer review of "Immunogenicity of a 20-Valent Pneumococcal Conjugate Vaccine Versus a 13-Valent Vaccine in Infants: A Systematic Review and Meta-Analysis"

_vaccines, 2025, doi:10.3390/vaccines13111156_

Round 1

Reviewer 1 Report

Comments and Suggestions for Authors
  1. The title is misleading: it should include 'in infants' because that is the focus of this meta analysis.
  2. In the Abstract the Results section is too limited. In the Conclusions it is said that WHO non-inferiority criteria are met, but that has not been studied in this meta analysis. So either redo that in your study or do not mention it in the Abstract.
  3. in the Methods section pg 3 different cutoffs for serotypes 5, 6B, 19A are mentioned. Please mention the reference that supports that deviation from the WHO definitions.
  4. some typos: toodler, 'common' serotypes should be 'shared' serotypes, opsonofago.. should be opsonophago..
  5. what is DP? It is used 5 times without explanation and I can only find PD later on.
  6. pg 6: for all eight serotypes... I do  not understand where this comes from.
  7. in Fig 3: line missing?
  8. in Fig 5: what happened to the confidence interval of some of the serotypes? Similarly: check range in e.g. serotype 23F and 10A.
  9. Pg 8. The difference in proportions reaching the predefined cutoffs is called 'smaller' in PCV20. If we look at Fig 5 the difference is 0. Why is it called smaller?
  10. It should be better explained why the pooled difference in protection level for the PCV20 unique serotypes is so close to protection levels in PCV13.
  11. Fig 6: in the figure it is better to write: 'PCV20 unique serotypes in PCV13 group' instead of 'Additional serotypes PCV13'.
  12. pooled OPA: I wonder if you can report pooled OPA using both 2 and 3 doses in primary series (see comment later). I think you can only pool the 3 studies with 3 doses and compare that with 3 doses of PCV13.
  13. In the Introduction and  Discussion you should better explain whether PCV20 unique serotypes (and which) are major pathogens in children and whether the protection in the meta analysis for those pathogens could be achieved. That may justify the switch from PCV13 to PCV20.

my biggest concern is that 2+1 cannot be compared to 3+1 in the difference in seroprotection after primary series and after the booster and not in the OPA. For the ratio, it may not be a problem. So the Results section should start with the statistical comparison of the 2+1 study and the pooled results of the 3+1 study (after primary series and after the booster). Some countries have changed their PCV13 schedule to 2+1 and replacing it with an immunogenically 'inferior' PCV20 2+1 may leave too many infants not protected (with too low antibody levels). 

That is something that should be evaluated too in the Discussion session: can you recommend a 2+1 schedule with PCV20 or should it preferably be a 3+1?

  1.  

Author Response

Comments 1: [The title is misleading: it should include 'in infants' because that is the focus of this meta analysis.]

Response 2: [Thank you for reviewer´s comment. As suggested, we have included “infants” in the title as follows:

Immunogenicity of a 20-valent pneumococcal conjugate vaccine versus a 13-valent vaccine in infants: a systematic review and meta-analysis”]

Comments 2: [In the Abstract the Results section is too limited. In the Conclusions it is said that WHO non-inferiority criteria are met, but that has not been studied in this meta analysis. So either redo that in your study or do not mention it in the Abstract.]

Response 2: [The reviewer’s comment seems judicious. As suggested, the abstract's results section as follows:

[…]Results: Four studies (4093 infants aged 42-180 days) were included in this meta-analysis which compared PCV20 and PCV13 vaccines published between 2021 and 2024. The immunogenicity of both groups was compared one month after the primary series and one month after the booster dose. The pooled results indicated that, compared with PCV13, PCV20 exhibited significantly reduced immunogenicity for shared serotypes after both the primary series and booster dose. However, immune responses were generally greater for the additional serotypes unique to PCV20. […]”

In addition, the sentences relating to the WHO non-inferiority criteria have been excluded from the abstract, as these were not analyzed in the present meta-analysis.]

Comments 3:  [In the Methods section pg 3 different cutoffs for serotypes 5, 6B, 19A are mentioned. Please mention the reference that supports that deviation from the WHO definitions.]

Response 3: [Thank you for the editor's comment. As suggested, we have included the reference, as indicated below:

 “[…]For the 13 matched serotypes, the PCV20 group was compared with the corresponding serotypes in the PCV13 group. For the seven additional serotypes, the PCV20 group was compared with the lowest result among the 13 serotypes in the PCV13 group (ex-cluding serotype 3 because of its atypical immunogenicity). The predefined IgG con-centrations were ≥0.35 μg/mL, except for serotypes 5 (≥0.23 μg/mL), 6B (≥0.10 μg/mL) and 19A (≥0.12 μg/mL) [3,10,41]. […]

Comments 4: [Some typos: toodler, 'common' serotypes should be 'shared' serotypes, opsonofago.. should be opsonophago..]

Response 4: [Thank you for reviewer´s comment. As suggested, we have replaced the term "common" to "shared." In addition, we have corrected all typos, including `opsonofagocytic´ and `toodler´.]

Comments 5: [what is DP? It is used 5 times without explanation and I can only find PD later on.]

Response 5: [Thank you for reviewer´s comment. The term "DP" is an abbreviation for "difference in percentage difference" (Difference in the percentage of participants who achieved predefined antibody levels for each serotype (PCV20 - PCV13)). To avoid any potential confusion, we have made the necessary changes. In addition, we have defined DP and PD in the abbreviation section of the manuscript.]

Comments 6: [pg 6: for all eight serotypes... I do not understand where this comes from.]

Response 6: [The reviewer’s comment seems judicious. We apologize for the typographical error; the correct number is seven instead of eight. We have corrected this concern in the main text as follows:

[…]As shown in Figure 2, there is a statistically significant reduction in the immunogenicity of PCV20 relative to that of PCV13 for all the shared serotypes. Among the additional serotypes, PCV20 had greater effects than PCV13 did for all seven serotypes, although these effects reached statistical significance only for serotypes 15B, 22F and 33F. […]”]

Comments 7: [in Fig 3: line missing?]

Response 7: [Thank you for reviewer´s comment. Figure 3 has been updated to ensure the image is accurate. We believe that the issue has now been resolved.]

Comments 8: [in Fig 5: what happened to the confidence interval of some of the serotypes? Similarly: check range in e.g. serotype 23F and 10A.]

Response 8: [The reviewer’s comment seems judicious. A comprehensive review of the data collected for the analysis in Figure 5 was conducted, followed by a re-examination of the meta-analysis. This process successfully resolved the issue, generating reliable confidence intervals for the serotypes 23F and 10A.]

Comments 9: [Pg 8. The difference in proportions reaching the predefined cutoffs is called 'smaller' in PCV20. If we look at Fig 5 the difference is 0. Why is it called smaller?]

Response 9: [Thank you for reviewer´s comment. While the difference in the percentage of participants who reached the predefined antibody value was similar between the two groups, it is important to note that the values were lower in the PCV20 group than in the PCV13 group for serotypes 1, 3, 4, 6A, 6B, 7F, 9V, 14, and 23F. We have replaced the term “smaller” to “lower” to improve understanding as follows:

[…] For the DP, Figure 5 shows that the effect was lower with PCV20 for most shared serotypes (1, 3, 4, 6A, 6B, 7F, 9V, 14 and 23F), whereas for serotypes 5, 18C, 19A and 19F, the effect was greater. […]”]

Comments 10: [It should be better explained why the pooled difference in protection level for the PCV20 unique serotypes is so close to protection levels in PCV13]

Response 10: [The reviewer’s comment seems judicious. As suggested, we have the following information on this concern:

For serotypes unique to PCV20, the results indicated that for serotypes 8, 10A, 11A, 15B, 22F and 33F, the percentage of participants that exceeded the prespecified GMC levels in the PCV20 group was greater than that in the PCV13 group. Only for serotype 12F was the result greater for PCV13. The results were statistically significant for serotypes 8 and 10A. Overall, the comparison between PCV20 and PCV13 indicates minimal differences. It should be noted that for the seven additional serotypes, the PCV20 group was compared with the lowest value among the 13 serotypes in the PCV13 group, which may explain the similarity in the observed values. [...]”]

Comments 11: [Fig 6: in the figure it is better to write: 'PCV20 unique serotypes in PCV13 group' instead of 'Additional serotypes PCV13'.]

Response11: [Thank you for reviewer´s comment. We have changed the sentence as suggested.

Comments 12: [pooled OPA: I wonder if you can report pooled OPA using both 2 and 3 doses in primary series (see comment later). I think you can only pool the 3 studies with 3 doses and compare that with 3 doses of PCV13.]

Response 12: [The reviewer’s comment seems judicious. In order to address this concern, we have conducted a subgroup analysis according to the vaccination schedule employed (3+1 or 2+1). These results are available in the supplementary material (tables S5 and S6). Furthermore, we have included the following information in the main text as follows:

In the Statistical analysis section:

            “[...]. 2.6. Data synthesis and statistical analysis

We conducted a random-effects meta-analysis via the Hartung-Knapp Sidik-Jonkman [43,44] method. We conducted subgroup analyses according  to the serotype and to the vaccination schedule employed (3+1 or 2+1), and estimated (i) pooled GMR, (ii) pooled DP, and (iii) pooled GMTs of OPA for PCV20 compared with those for PCV13 with 95% confidence intervals (95% CIs).[...]”

In the result section:

            “[...]. The results of the subgroup analysis according to the type of vaccination schedule (3+1 or 2+1) for each serotype are shown in supplementary tables S5 and S6.[...]”]

Comments 13: [In the Introduction and Discussion you should better explain whether PCV20 unique serotypes (and which) are major pathogens in children and whether the protection in the meta analysis for those pathogens could be achieved. That may justify the switch from PCV13 to PCV20.

my biggest concern is that 2+1 cannot be compared to 3+1 in the difference in seroprotection after primary series and after the booster and not in the OPA. For the ratio, it may not be a problem. So the Results section should start with the statistical comparison of the 2+1 study and the pooled results of the 3+1 study (after primary

series and after the booster). Some countries have changed their PCV13 schedule to 2+1 and replacing it with an immunogenically 'inferior' PCV20 2+1 may leave too many infants not protected (with too low antibody levels).

That is something that should be evaluated too in the Discussion session: can you recommend a 2+1 schedule with PCV20 or should it preferably be a 3+1?]

 Response 13: [Thank you for reviewer´s comment. As suggested, we have explained more clearly the involvement of the unique serotypes of PCV20 in invasive pneumococcal disease, and why this could rationalize the transition from PCV13 to PCV20:

In the introduction section:

However, given that some serotypes not covered by current vaccines continue to cause significant disease, and that a considerable burden of IPD persists in children, with infants under one year of age being particularly vulnerable, there is an urgent need for higher valence pneumococcal vaccines covering an increasing number of S. pneumoniae serotypes [8]. For example, in Europe, an increasing proportion of IPD cases are caused by serotypes unique to PCV20, accounting for more than 61.8% of all IPD cases among children aged less than 1 year in 2022 [20]. [...]”

In the discussion section:

“[...]. Although PCV20 showed lower immunogenicity for shared serotypes, it met the WHO criteria for noninferiority, including GMC and the proportion of infants reaching predefined antibody thresholds [3,10,25,75]. It is unknown whether a lower serotype-specific GMC of antibody indicates less efficacy for those serotypes. The threshold is meant to be used to establish noninferiority against the reference PCV in aggregate, and no serotype-specific thresholds have been defined [3]. Nevertheless, noninferiority does not imply clinical equivalence. The sequential use of noninferiority criteria may have cumulative negative consequences, and even modest reductions in antibody re-sponses could translate into diminished protection over time, particularly in popula-tions with lower vaccine responsiveness or population suboptimal coverage [76]. However, lower antibody titers do not necessarily indicate lower clinical efficacy. The WHO position paper states that there is no established correlation of protection for many serotypes, and historical data show that PCVs substantially reduce IPD even when antibody levels are lower. Future studies should clarify whether these differences in immunogenicity correlate with differences in vaccine effectiveness, especially with respect to protection against IPD [77].  [...]

To assess whether a 3+1 vaccination schedule is more advantageous than a 2+1 schedule, the OPA results for the 3+1 schedule are considerably higher than those for the 2+1 schedule. This suggests that the 3+1 schedule may offer enhanced protection. However, the available evidence is insufficient to draw a conclusion and make a deci-sion. This is because the results depend on several factors. These include the immuno-genicity results obtained, the income of the country where the schedule is implement-ed, the circulating serotypes, the incidence of pneumococcal disease, and other cost-effectiveness analyses in each specific country. It is evident that further studies are necessary to thoroughly analyze these factors and obtain additional evidence.

Implications for practice and policy

While the use of PCV20 appears justified on the basis of WHO criteria and broader serotype coverage, a titer antibody that may correlate with protection against IPD caused by any one serotype is unknown [77]. Therefore, national immunization programs should consider the potential trade-off between reduced immunogenicity for shared serotypes and expanded protection against emerging serotypes. Epidemiological surveillance of circulating serotypes and cost-effectiveness analyses and studies should be integrated with immunogenicity findings to inform public health decisions, particularly in low- and middle-income countries where disease burden and vaccine affordability vary substantially [80].

Reviewer 2 Report

Comments and Suggestions for Authors

Estimated Authors,

I've read with interest the present paper entitled "Immunogenicity of a 20-valent pneumococcal conjugate vaccine versus a 13-valent vaccine: a systematic review and meta-analysis". The present study, by means of a solid and rigorous approach provides an estimate of immunogenicity of PCV20 vs. PCV13 through available RCTs. 

The study is of certain interest due to the increasing doubts about the improved effectiveness of PCV20 compared to PCV13 in real world settings. Even though the present paper does not deal with efficacy nor effectiveness, but only with immunogenicity, the provided information can be useful for health authorities involved into the choice of future vaccination strategies.

Eventually, Pacheco-Haro et al provide a significant message: PCV20 compared to PCV13 can be less effective in eliciting serum antibody levels, but also guarantees noticeable antibody levels on non-PCV13 serotypes.

The article is well written, and accurately drafted in terms of study design and application of up-to-date methodology. The implicit limits of the paper have been accurately described by Study Authors in the Limits sections. Honestly, the present reviewer has no recommendations for likely improvement of a paper whose acceptance nearly "as it is" I'm therefore endorsing. In fact, the only suggestion I can share with study Authors is stressing in discussion section how the correlate between immunogenicity and effectiveness is not so strightforward (see for example: https://pmc.ncbi.nlm.nih.gov/articles/PMC9640717/). Moreover, immunogenicity does not predict how the vaccination can impair the circulation of the pathogen in the general population, stressing in a balance of pros and cons the potential role of vaccines with more limited immunogenicity but increased capability of impacting on the circulation of the pathogen in the general population through mucosal immunity.

Author Response

Comments 1: [I've read with interest the present paper entitled "Immunogenicity of a 20-valent pneumococcal conjugate vaccine versus a 13-valent vaccine: a systematic review and meta-analysis". The present study, by means of a solid and rigorous approach provides an estimate of immunogenicity of PCV20 vs. PCV13 through available RCTs.

The study is of certain interest due to the increasing doubts about the improved effectiveness of PCV20 compared to PCV13 in real world settings. Even though the present paper does not deal with efficacy nor effectiveness, but only with immunogenicity, the provided information can be useful for health authorities involved into the choice of future vaccination strategies.

Eventually, Pacheco-Haro et al provide a significant message: PCV20 compared to PCV13 can be less effective in eliciting serum antibody levels, but also guarantees noticeable antibody levels on non-PCV13 serotypes.

The article is well written, and accurately drafted in terms of study design and application of up-to-date methodology. The implicit limits of the paper have been accurately described by Study Authors in the Limits sections. Honestly, the present reviewer has no recommendations for likely improvement of a paper whose acceptance nearly "as it is" I'm therefore endorsing. In fact, the only suggestion I can share with study Authors is stressing in discussion section how the correlate between immunogenicity and effectiveness is not so strightforward (see for example: https://pmc.ncbi.nlm.nih.gov/articles/PMC9640717/). Moreover, immunogenicity does not predict how the vaccination can impair the circulation of the pathogen in the general population, stressing in a balance of pros and cons the potential role of vaccines with more limited immunogenicity but increased capability of impacting on the circulation of the pathogen in the general population through mucosal immunity.]

Response 1: [We would like to express our sincere gratitude for the time and effort you devoted to reviewing our manuscript entitled “Immunogenicity of a 20-valent pneumococcal conjugate vaccine versus a 13-valent vaccine: a systematic review and meta-analysis.”

We sincerely thank you for the thoughtful and encouraging comments on our manuscript. We are very pleased to know that the study design, methodology, and overall presentation were appreciated.

We fully agree with your insightful observation that the relationship between immunogenicity and real-world effectiveness is not straightforward. As suggested, we have revised the Discussion section to further emphasize this point and have included the reference indicated (https://pmc.ncbi.nlm.nih.gov/articles/PMC9640717/) to support this consideration, as follows on page 13:

 “[...]. Implications for practice and policy

While the use of PCV20 appears justified on the basis of WHO criteria and broader serotype coverage, a titer antibody that may correlate with protection against IPD caused by any one serotype is unknown [77]. Linking immunogenicity and effectiveness is necessary to assess whether changes in immune response from currently recommended PCVs to next-generation vaccines could impact effectiveness [80]. Therefore, national immunization programs should consider the potential trade-off between reduced immunogenicity for shared serotypes and expanded protection against emerging serotypes. Epidemiological surveillance of circulating serotypes, and effectiveness and cost-effectiveness analyses and studies should be integrated with immunogenicity findings to inform public health decisions, particularly in low- and middle-income countries where the disease burden and vaccine affordability vary substantially [81].[...]”]

Round 2

Reviewer 1 Report

Comments and Suggestions for Authors

Good job!

  1. Done
  2. better specify what is meant with 'The pooled results indicated that, compared with PCV13, PCV20 exhibited significantly reduced immunogenicity for shared serotypes', because you explain before that you focus on 3 outcomes (GMR, PD, OPA titers). Which of these is significantly reduced for most shared serotypes? In my opinion GMRs most because PD remained the same?
  3. Done
  4. Done
  5. DP is done now
  6. done
  7. done
  8. just check 7F. Is the upper limit -0.33 or +0.33? Check 23 F, too. And 15B: -0.00? Why not +0.00? Delete the '-'.
  9. Excuse me, but I do not see it: there is no 'lower' effect (bad word). And no greater effect for 5, 18c, 19a, 19f. The confidence interval always includes zero in Fig 5. Consequently, it is similar and not 'smaller' or 'lower'? What do you mean: 'Although the pooled GMRs were always significantly lower for the shared serotypes, the proportions above the prespecified cutoff titres for the shared serotypes in both groups were not significantly different'. Then you can understand that this is in line with the WHO non-inferiority claim? So, rephrase or explain me better what you mean, because the proportions are similar. Also useful suggestion for the Abstract?
  10. For me it is not clear what is now written "the PCV20 group was compared with the lowest value among the 13 serotypes". Which 'value'?
  11. done
  12. could not find suppl table S5 and S6, but I would like to see a small section in the beginning of Results what is the result of this subgroup analysis. Is the 3+1 schedule more immunogenic than 2+1, or whether your analysis could not make that conclusion.
  13. You have kept the pooled data in table 6 and 7 regardless of schedule (3+1 or 2+1). I suggest you just provide the 3+1 (most studies) and put 2+1 in Supplementary data. This is better, because OPA after 2+1 is always different than those in 3+1. Introduction part: is now changed what good additions to the text. Later this new sentence can be deleted because you already explained it at length "However, lower antibody titers do not necessarily indicate lower clinical efficacy. The WHO position paper states that there is no established correlation of protection for many serotypes, and historical data show 
    that PCVs substantially reduce IPD even when antibody levels are lower." Furthermore, please shorten the part later about 'To assess whether a 3+1 vaccination schedule is more advantageous than a 2+1 schedule, the OPA results for the 3+1 schedule are considerably higher than those for the 2+1 schedule. This suggests that the 3+1 schedule may offer enhanced protection. However, the available evidence is insufficient to draw a conclusion and make a deci-sion. This is because the results depend on several factors. These include the immuno-genicity results obtained, the income of the country where the schedule is implement-ed, the circulating serotypes, the incidence of pneumococcal disease, and other cost-effectiveness analyses in each specific country. It is evident that further studies are necessary to thoroughly analyze these factors and obtain additional evidence.' This is a limitation of the data so far, but most countries have moved to 2+1 due to the fact immunogenicity after the booster dose following a two-dose primary schedule was satisfactory, with effectiveness against invasive pneumococcal disease (IPD) maintained, compared to 3+1. The UK even moved to 1+1 with PCV13. Subsequently, just explain that although 'immunogenicity of the 3+1 was better when assessing the OPA outcomes compared to 2+1. This is in line with other findings Pneumococcal conjugate vaccine schedule: 3+1, 2+1, or 1+1? - The Lancet Child & Adolescent Health, without compromising vaccine effectiveness against IPD in children.' However, with the lower GMTs for PCV20 compared to PCV13 additional reductions in the number of childhood pneumococcal vaccine, such as the 1+1 schedule in the UK, may ultimately results in reduced effectiveness. This requires surveillance etc. Just reduce the lengthy explanation, please.

Author Response

Comments 1: [better specify what is meant with 'The pooled results indicated that, compared with PCV13, PCV20 exhibited significantly reduced immunogenicity for shared serotypes', because you explain before that you focus on 3 outcomes (GMR, PD, OPA titers). Which of these is significantly reduced for most shared serotypes? In my opinion GMRs most because PD remained the same?]

Response 1: [The reviewer’s comment seems judicious. As suggested, the abstract's results section is as follows:

'[...]. Abstract

[...]. Results: Four studies (4093 infants aged 42–180 days) that compared the PCV20 and PCV13 vaccines, published between 2021 and 2024, were included in this me-ta-analysis.The immunogenicity of both groups was compared one month after the primary series and one month after the booster dose. The pooled results indicated that PCV20 elicited lower immune responses for the 13 serotypes shared with PCV13, ac-cording to the GMR and OPA outcomes. For the DP outcome, no statistically signifi-cant differences were observed between the two groups. Immune responses were were higher for the additional serotypes in the PCV20 group; however, these differences were not statistically significant for all serotypes. Discussion: This meta-analysis of-fers an overview of the evidence on the comparative immunogenicity of PCV20 and PCV13. Although some outcomes indicate that PCV20 elicits lower immune responses for the 13 serotypes shared with PCV13, it provides immunity against seven additional serotypes associated with IPD. Further studies are warranted to strengthen the evi-dence base, and continuous IPD surveillance remains essential to monitor shifts in serotype prevalence, assess the impact of current and future vaccines, and guide vac-cine policy recommendations.[...]

[...]. For the DP outcome, Figure 5 shows that no statistically significant differences were observed between the two groups for the shared serotypes. Only the results for serotypes 3 and 23F were statistically significant.[...]

[...] 4. Discussion

This meta-analysis synthesizes the available evidence on the immunogenicity of PCV20 compared with that of PCV13 in healthy infants. These findings indicate that PCV20 exhibits reduced immunogenicity for the 13 serotypes shared with PCV13, based on the analyzed outcomes, except for the DP outcome, for wich no statistically significant differences were observed between the two groups. However, PCV20 confers immunogenicity against seven additional serotypes (8, 10A, 11A, 12F, 15B, 22F, and 33F), which are not covered by PCV13 and are increasingly associated with IPD [29,31,48–51]. [...].

[...]. Although PCV20 showed reduced immunogenicity for the shared serotypes in the outcomes of the GMR and OPA in this meta-analysis, it met the WHO criteria for noninferiority (the percentage of PCV recipients with serotype-specific immunoglobulin G ≥0.35 μg/mL and the serotype-specific IgG geometric mean concentration after vaccination). [3,10,25,75]. [...].”

[...] 5. Conclusions

This systematic review and meta-analysis provides the first comprehensive synthesis of comparative immunogenicity data between PCV20 and PCV13 in healthy infants. Although PCV20 elicited lower immune responses for the 13 shared serotypes in some outcomes, [...]´]

Comments 2: [ just check 7F. Is the upper limit -0.33 or +0.33? Check 23 F, too. And 15B: -0.00? Why not +0.00? Delete the '-'.]

Response 2: [The reviewer’s comment seems judicious. To verify the results discussed, a comprehensive review of the data collected for the analysis in Figure 5 was conducted, followed by a re-examination of the meta-analysis. . The intervals corresponding to the following serotypes are shown below:

Serotype 7F: -0.39 [ -0.46; -0.33]

Serotype 23F: -0.87 [ -1.46; -0.28]

Serotype 15B: 2.45 [-0.002; 4.91]

We would like to clarify that this value was rounded to two decimal places. As mentioned previously, the exact value to three decimal places is -0.002. This correction has also been reflected in the forest plot.]

Comments 3: [Excuse me, but I do not see it: there is no 'lower' effect (bad word). And no greater effect for 5, 18c, 19a, 19f. The confidence interval always includes zero in Fig 5. Consequently, it is similar and not 'smaller' or 'lower'? What do you mean: 'Although the pooled GMRs were always significantly lower for the shared serotypes, the proportions above the prespecified cutoff titres for the shared serotypes in both groups were not significantly different'. Then you can understand that this is in line with the WHO non-inferiority claim? So, rephrase or explain me better what you mean, because the proportions are similar. Also useful suggestion for the Abstract?]

Response 3: [Thank you for your comment. The text has been modified to facilitate comprehension of the results obtained, as follows:

[…]. 3.3. Meta-analysis

[...]For the DP outcome, Figure 5 shows that no statistically significant differences were observed between the two groups for the shared serotypes. Only the results for serotypes 3 and 23F were statistically significant. […]

“[…]. 4. Discussion

[…]. Although PCV20 showed reduced immunogenicity for the shared serotypes in the outcomes of the GMR and OPA in this meta-analysis, it met the WHO criteria for noninferiority (the percentage of PCV recipients with serotype-specific immunoglobulin G ≥0.35 μg/mL and the serotype-specific IgG geometric mean concentration after vaccination). [3,10,25,75]. […]]

Comments 4: [For me it is not clear what is now written "the PCV20 group was compared with the lowest value among the 13 serotypes". Which 'value'?]

Response 4: [Thank you for the reviewer´s comment: "the PCV20 group was compared with the lowest value among the 13 serotypes" is an explanation of the origin of the results we analyzed, which are already provided directly by each study. Please note that the original data from each of the studies are available for review.

For example, the GMR outcome after the booster dose in Korbal et al, 2024 Studio for serotype 8 is calculated based on the GMC of the PCV20 group (3.57) and that of the PCV13 group (2.41), which represents the lowest GMC value among the shared serotypes.

Rather than presenting each individual value, which are reported in the original studies, our analysis aims to synthesize the results and highlight overall trends across the studies.]

Comments 5: [could not find suppl table S5 and S6, but I would like to see a small section in the beginning of Results what is the result of this subgroup analysis. Is the 3+1 schedule more immunogenic than 2+1, or whether your analysis could not make that conclusion.]

Response 5: [The reviewer’s comment seems judicious. To address this concern, we have added a small section to the Results section as follows:

“[...]. The results of the subgroup analysis according to vaccination schedule type (3+1 or 2+1) for each serotype are presented in Supplementary Tables S5 and S6. The immunogenicity results derived from the subgroup analysis indicate that the 3+1 schedule elicited a more robust immune response than the 2+1 schedule for all serotypes, both after the primary series and following to the booster dose. However, the available evidence is limited, as the analysis included only a single study using the 2+1 schedule. [...]”]

Comments 6: [You have kept the pooled data in table 6 and 7 regardless of schedule (3+1 or 2+1). I suggest you just provide the 3+1 (most studies) and put 2+1 in Supplementary data. This is better, because OPA after 2+1 is always different than those in 3+1. Introduction part: is now changed what good additions to the text. Later this new sentence can be deleted because you already explained it at length "However, lower antibody titers do not necessarily indicate lower clinical efficacy. The WHO position paper states that there is no established correlation of protection for many serotypes, and historical data show that PCVs substantially reduce IPD even when antibody levels are lower." Furthermore, please shorten the part later about 'To assess whether a 3+1 vaccination schedule is more advantageous than a 2+1 schedule, the OPA results for the 3+1 schedule are considerably higher than those for the 2+1 schedule. This suggests that the 3+1 schedule may offer enhanced protection. However, the available evidence is insufficient to draw a conclusion and make a decision. This is because the results depend on several factors. These include the immuno-genicity results obtained, the income of the country where the schedule is implemented, the circulating serotypes, the incidence of pneumococcal disease, and other cost-effectiveness analyses in each specific country. It is evident that further studies are necessary to thoroughly analyze these factors and obtain additional evidence.' This is a limitation of the data so far, but most countries have moved to 2+1 due to the fact immunogenicity after the booster dose following a two-dose primary schedule was satisfactory, with effectiveness against invasive pneumococcal disease (IPD) maintained, compared to 3+1. The UK even moved to 1+1 with PCV13. Subsequently, just explain that although 'immunogenicity of the 3+1 was better when assessing the OPA outcomes compared to 2+1. This is in line with other findings Pneumococcal conjugate vaccine schedule: 3+1, 2+1, or 1+1? - The Lancet Child & Adolescent Health, without compromising vaccine effectiveness against IPD in children.' However, with the lower GMTs for PCV20 compared to PCV13 additional reductions in the number of childhood pneumococcal vaccine, such as the 1+1 schedule in the UK, may ultimately results in reduced effectiveness. This requires surveillance etc. Just reduce the lengthy explanation, please.]

Response 6: [Thank you for your comment. As suggested, Figures 6 and 7 have been replaced with meta-analyses corresponding to the analysis of the study group with the 3+1 schedule. Please refer to Tables S5 and S6 in the supplementary material, where the results of the analysis are shown separately for each 3+1 or 2+1 vaccination schedule:

“[...]Figures 6 and 7 present the pooled results from the 3+1 vaccination studies. Tables S5 and S6 in the supplementary material show the results of for the 2+1 vaccination schedule study.[...]”

As suggested, this sentence has been deleted: "However, lower antibody titers do not necessarily indicate lower clinical efficacy. The WHO position paper states that there is no established correlation of protection for many serotypes, and historical data show that PCVs substantially reduce IPD even when antibody levels are lower."  

The text of the discussion regarding the vaccination results according to the vaccination schedule has been revised in accordance with the recommendations provided, resulting in the following revised text:

“[....]. Compared with the 2+1 schedule, the 3+1 vaccination schedule has been shown to elicit higher OPA responses. However, some findings [80] indicate that these differ-ences do not reduce vaccine efficacy against IPD in children. Further reductions in the number of childhood pneumococcal vaccines, such as the 1+1 schedule implemented in the United Kingdom, may ultimately result in reduced effectiveness. Additional re-search is needed to determine the overall superiority of each schedule. When form-lating a schedule, it is essential to consider factors such as the country´s incomelevel, prevalent serotypes, incidence of pneumococcal disease, and cost-effectiveness analyses. [...]”]
